# End-to-End Meta-Bayesian Optimisation with Transformer Neural Processes

**Alexandre Maraval**[*]
Huawei Noah's Ark Lab
alexandre.maraval1@huawei.com

**Matthieu Zimmer**[*]
Huawei Noah's Ark Lab
matthieu.zimmer@huawei.com

**Antoine Grosnit**
Huawei Noah's Ark Lab
Technische Universität Darmstadt
antoine.grosnit2@huawei.com

**Haitham Bou Ammar**
Huawei Noah's Ark Lab,
University College London
haitham.ammar@huawei.com

## Abstract

Meta-Bayesian optimisation (meta-BO) aims to improve the sample efficiency of Bayesian optimisation by leveraging data from related tasks. While previous methods successfully meta-learn either a surrogate model or an acquisition function independently, joint training of both components remains an open challenge. This paper proposes the first end-to-end differentiable meta-BO framework that generalises neural processes to learn acquisition functions via transformer architectures. We enable this end-to-end framework with reinforcement learning (RL) to tackle the lack of labelled acquisition data. Early on, we notice that training transformer-based neural processes from scratch with RL is challenging due to insufficient supervision, especially when rewards are sparse. We formalise this claim with a combinatorial analysis showing that the widely used notion of regret as a reward signal exhibits a logarithmic sparsity pattern in trajectory lengths. To tackle this problem, we augment the RL objective with an auxiliary task that guides part of the architecture to learn a valid probabilistic model as an inductive bias. We demonstrate that our method achieves state-of-the-art regret results against various baselines in experiments on standard hyperparameter optimisation tasks and also outperforms others in the real-world problems of mixed-integer programming tuning, antibody design, and logic synthesis for electronic design automation.

## 1 Introduction

Bayesian optimisation (BO) techniques are sample-efficient sequential model-based solvers optimising expensive-to-evaluate black-box objectives. Traditionally, BO methods operate in isolation focusing on one task at a time, a setting that led to numerous successful applications, including but not limited to hyperparameter tuning [1], drug and chip design [2], and robotics [3]. Although successful, focusing on only one objective in isolation may increase sample complexities on new tasks since standard BO algorithms work tabula-rasa as they start optimising from scratch, ignoring previously observed black-box functions.

To improve sample efficiency on new *target* tasks, meta-BO makes use of data collected on related *source* tasks [4] and attempts to transfer knowledge in between. Those methods are primarily composed of two parts: 1) a (meta) surrogate model predicting the function to optimise and 2) a (meta) acquisition function (AF) estimating how good a queried location is. Regarding surrogate

---

[*]These authors contributed equally to this work

37th Conference on Neural Information Processing Systems (NeurIPS 2023).

modelling, prior work relies on Gaussian processes (GPs) to perform transfer across tasks due to their data efficiency [5–9], or focuses on deep neural networks [10–12] gaining from their representation flexibility. As for acquisition functions, previous works assumed a fixed GP and trained a neural network to perform transfer [13, 14].

Although successful, those methods rely on the assumption that the two steps described above can be handled independently, potentially missing benefits from considering both steps together. Therefore, this paper advocates for an end-to-end training protocol for meta-BO where we train a model predicting AF values directly from observed data, without relying on a GP surrogate.

As shown in [10, 12, 15, 16], Neural processes (NP) are a good candidate for meta-learning due to their structural properties, thus we propose to use a new transformer-based NP to model the acquisition function directly [17].

Nevertheless, the lack of labelled AF data prohibits a supervised learning protocol. Here, we rely on reward functions to assess the goodness of AFs and formulate a new reinforcement learning (RL) problem. Our RL formulation attempts to learn an optimal policy that selects new evaluation probes by minimising per-task cumulative regrets.

Early on, we faced very unstable training with minimal learning due to the combination of RL with transformers [18]. Furthermore, we notice that this problem amplifies when the reward function is sparse, which is the case in our setting, as we formally show in Section 3.2. To mitigate this difficulty, we introduce an inductive bias in our transformer architecture via an auxiliary loss [19] that guides a part of the network to learn a valid probabilistic model, effectively specialising as a neural process with gradient updates. Having developed our framework, we demonstrate state-of-the-art regret performance in hyperparameter optimisation of real-world problems and combinatorial sequence optimisation for antibody and chip design (see Section 4). Those solid empirical results show the value of end-to-end training in meta-BO to further improve sample complexity.

**Contributions** We summarise our contributions as follows: *i)* developing the first end-to-end transformer-based architecture for meta-BO predicting acquisition function values, *ii)* identifying logarithmic reward sparsity patterns hindering end-to-end learning with RL, *iii)* introducing NP-based inductive biases for successful model updates, and *iv)* demonstrating new state-of-the-art regret results on a wide range of traditional (hyperparameter tuning) and non-traditional (chip and antibody design) real-world benchmarks.

## 2 Background

### 2.1 Bayesian Optimisation

In BO, we sequentially maximise an expensive-to-evaluate black-box function $f(\boldsymbol{x})$ for variables $\boldsymbol{x} \in \mathcal{X}$. BO techniques operate in two steps. First, based on previously collected evaluation data, we fit a probabilistic surrogate model to emulate $f(\boldsymbol{x})$ allowing us to make probabilistic predictions of the function's behaviour on unobserved input points. Given the first step's probabilistic predictions, the second step in BO optimises an AF that trades off exploration and exploitation to find a new input query, $\boldsymbol{x}_{\text{new}} \in \mathcal{X}$, for evaluation. Both the model and acquisition choices play a critical role in the success of BO. Many works adopt GPs as surrogate models due to their sample efficiency and practical uncertainty estimation [20]. Moreover, on the acquisition side, the common practice is to optimise one from a set of widely adopted AFs such as Expected Improvement (EI) [21].

### 2.2 Transfer in Bayesian Optimisation

Transfer techniques in BO [4, 13] aim to improve the optimisation of a newly observed *target* black-box function by reusing knowledge from past experiences gathered in related *source* domains, i.e. solve $\max_{\boldsymbol{x} \in \mathcal{X}} f_{\text{Target}}(\boldsymbol{x})$ by leveraging information from $K$ source black-boxes $f_1(\boldsymbol{x}), \ldots, f_K(\boldsymbol{x})$. We assume this information is available to our agent via $\mathcal{D}_1, \ldots, \mathcal{D}_K$ such that each dataset $\mathcal{D}_k = \{\langle \boldsymbol{x}_i^{(k)}, y_i^{(k)} \rangle\}_{i=1}^{n^{(k)}}$ consists of $n^{(k)}$ (noisy) evaluations of $f_k(\boldsymbol{x})$ for all $k \in [1:K]$. Following the meta-learning literature [4, 5, 13], we impose no additional assumptions on the process that collects $\mathcal{D}_1, \ldots, \mathcal{D}_K$, allowing for a diverse set of source optimisation algorithms, including but not limited to BO, genetic and evolutionary algorithms [22], and sampling-based strategies [23]. Many algorithms that use source data to improve performance on target domains exist. We categorise those based

on the part they customise within the BO pipeline, i.e., by registering if they affect initial points [24, 25], search spaces [26], surrogate models [5] or AFs [13, 14]. Since our work devises end-to-end meta-BO pipelines, we now elaborate on critical surrogate models needed for the rest of the paper and leave a detailed presentation of related work to Section 5.

Although GPs [20] are a prominent tool for single-task BO, high computational complexity [27], among other drawbacks (e.g. smoothness assumptions or limited scalability in terms of dimensions), can limit their application to transfer scenarios. Of course, one can generalise GPs to support transfer using multi-task kernels [28] or GP ensembles [6], for example. However, recent trends demonstrated superior performance to such extensions when using deep networks (e.g., neural processes) to meta-learn probabilistic surrogates [5, 13].

**Neural Processes** A popular method for effective meta-learning consists of directly inputting context points (i.e., available data to adjust to a target task) for a model to adapt to unseen tasks [15]. We can accomplish such a goal by relying on neural processes (NPs), a class of models that combine the flexibility of deep neural networks with the properties of stochastic processes [10]. Given an observation set of input-output pairs $\mathcal{D}_{\mathrm{obs}}$ and a set of $n_{\mathrm{pred}}$ locations $\boldsymbol{x}^{(\mathrm{pred})}$ at which we desire to make predictions, an NP parameterised by $\boldsymbol{\theta}$ outputs a distribution $p_{\boldsymbol{\theta}}(\cdot|\boldsymbol{x}^{(\mathrm{pred})}, \mathcal{D}_{\mathrm{obs}})$ that approximates the true posterior over the labels $y^{(\mathrm{pred})}$. Among the different parameterisations of $p_{\boldsymbol{\theta}}(\cdot)$, e.g., in [29–31], transformer architectures [17] recently emerged as compelling choices of NPs [11, 12]. In this work, we also adopt a transformer architecture inspired by this architecture's broad software support and state-of-the-art supervised learning results as reported in [11].

## 2.3   Reinforcement Learning

In Section 3, we attempt to learn an end-to-end model that predicts acquisition values. Of course, we cannot fit the parameters of such a model with supervised learning due to the lack of labelled acquisition data. RL [32] is a viable alternative for learning from delayed and non-differentiable reward signals in those cases. In RL, we formalise problems as Markov decision processes (MDP): $\mathcal{M} = \langle \mathcal{S}, \mathcal{A}, \mathcal{P}, \mathcal{R}, \gamma \rangle$, where $\mathcal{S}$ and $\mathcal{A}$ denote the state and action spaces, $\mathcal{P} : \mathcal{S} \times \mathcal{A} \times \mathcal{S} \rightarrow [0, 1]$ the state transition model, $\mathcal{R}$ the reward function dictating the optimisation goal, and $\gamma \in [0, 1)$ a discount factor specifying the degree to which rewards are discounted over time. A policy $\pi : \mathcal{S} \times \mathcal{A} \rightarrow [0, 1]$ is an action-selection rule that is defined as a probability distribution over state-action pairs, where $\pi(\boldsymbol{a}_t|\boldsymbol{s}_t)$ represents the probability of selecting action $\boldsymbol{a}_t$ in state $\boldsymbol{s}_t$. An RL agent aims to find an optimal policy $\pi^{\star}$ that maximises (discounted) expected returns. For determining $\pi^{\star}$, we use the state-of-the-art proximal policy optimisation (PPO) algorithm [33].

## 3   End-to-End Meta-Bayesian Optimisation

While transfer techniques in BO saw varying degrees of success in many applications, current approaches lack end-to-end differentiability, where model learning and acquisition discovery arise as two separate steps. Specifically, the surrogate model gradients hardly affect the acquisition network, and the acquisition's network gradients fail to back-propagate to the surrogate's updates.

Enabling end-to-end transfer frameworks in which we learn the surrogate and acquisition jointly hold the promise for more scalable and easier-to-deploy algorithms that are more robust to input data or task changes. Following such a framework, we also expect more accurate predictions that can lead to better regret results (see Section 4) since we optimise the entire transfer pipeline, including the intermediate probabilistic model and AF representations. Additionally, end-to-end training techniques allow us to mitigate the need for domain-specific expertise and permit stable implementations that benefit fully from GPU and computing hardware.

The most straightforward way to enable end-to-end training in Bayesian optimisation is to introduce a deep network that acquires search variables and historical evaluations of black-box functions as inputs and outputs acquisition values after a set of nonlinear transformations. Of course, it is challenging to fit the weights of such a network due, in part, to a lack of labelled acquisition data where search variables and history of evaluations are inputs and acquisition values are labels.

## 3.1 Reinforcement Learning for End-to-End Training

Our approach utilises RL to fit the network's parameters $\boldsymbol{\theta}$ from minimal supervision, circumventing the need for labelled acquisition data. To formalise the RL problem, we introduce an MDP where:

**State**: $\boldsymbol{s}_t = [\mathcal{H}_t, t, T]$ (history, BO time-step & budget)  **Action**: $\boldsymbol{a}_t = \boldsymbol{x}_t$ (choice of new probe),

with $\mathcal{H}_t = \{\langle \boldsymbol{x}_1, y_1 \rangle, \ldots, \langle \boldsymbol{x}_{t-1}, y_{t-1} \rangle\}$ denoting the history of black-box evaluations up-to the current time-step $t$. Adding the current BO step $t$ and the maximum budget $T$ in our state variable $\boldsymbol{s}_t$ helps balance exploration versus exploitation trade-offs as noted in [34]. Our MDP's transition function is straightforward, updating $\mathcal{H}_t$ by appending newly evaluated points, i.e., $\mathcal{H}_{t+1} = \mathcal{H}_t \cup \{\langle \boldsymbol{x}_t, y_t \rangle\}$ and incrementing the time variable $t$. Regarding rewards, we follow the well-established literature [13, 14] and define $r_t = \max_{1 \leq \ell \leq t} y_\ell$ to correspond to simple regret. Given such an MDP, our agent attempts to find a parameterised policy $\pi_{\boldsymbol{\theta}}$ which, when conditioned on $\boldsymbol{s}_t$, proposes a new probe $\boldsymbol{x}_t$ that minimises cumulative regret (i.e., the sum of total discounted simple regrets).

**Extensions to Multi-Task Reinforcement Learning** The above MDP describes an RL configuration that learns $\boldsymbol{\theta}$ in a single BO task. We now extend this formulation to multi-task scenarios allowing for a meta-learning setup. To do so, we introduce a set of MDPs $\mathcal{M}_1, \ldots, \mathcal{M}_K$ with $K$ being the total number of available tasks. This paper considers same-domain multi-task learning scenarios. As such, we assume that all MDPs share the same state and action spaces and leave cross-domain extensions as an interesting avenue for future work. We define each MDP, $\mathcal{M}_k$, as previously introduced such that:

**States:** $\boldsymbol{s}_t^{(k)} = [\mathcal{H}_t^{(k)}, t^{(k)}, T^{(k)}]$  **Actions:** $\boldsymbol{a}_t^{(k)} = \boldsymbol{x}_t^{(k)}$  **Rewards:** $r_t^{(k)} = \max_{1 \leq \ell \leq t^{(k)}} y_t^{(k)} \ \forall k.$

Moreover, for each task $k$, the transition model updates task-specific histories with $\mathcal{H}_{t+1}^{(k)} = \mathcal{H}_t^{(k)} \sqcup \{\langle \boldsymbol{x}_t^{(k)}, y_t^{(k)} \rangle\}$ and increments $t^{(k)}$. Contrary to the single task setup, we now seek a policy $\pi_{\boldsymbol{\theta}}$ which performs well on average across $K$ tasks:

$$\arg\max_{\pi_{\boldsymbol{\theta}}} J(\pi_{\boldsymbol{\theta}}) = \arg\max_{\pi_{\boldsymbol{\theta}}} \mathbb{E}_{k \sim p_{\text{tasks}}} \left[ \mathbb{E}_{\mathcal{H}_{T^{(k)}}^{(k)} \sim p_{\pi_{\boldsymbol{\theta}}}} \left[ \sum_{t=1}^{T^{(k)}} \gamma^{t-1} r_t^{(k)} \right] \right], \quad (1)$$

where $p_{\text{tasks}}$ denotes the task distribution. Furthermore, the per-task history distribution $p_{\pi_{\boldsymbol{\theta}}}(\mathcal{H}_{T^{(k)}}^{(k)})$ is jointly parameterised by $\boldsymbol{\theta}$ and defined as:

$$p_{\pi_{\boldsymbol{\theta}}} \left( \mathcal{H}_{T^{(k)}}^{(k)} \right) = p \left( y_{T^{(k)}-1}^{(k)} \Big| \boldsymbol{x}_{T^{(k)}-1}^{(k)} \right) \pi_{\boldsymbol{\theta}} \left( \boldsymbol{x}_{T^{(k)}-1}^{(k)} | \mathcal{H}_{T^{(k)}-1}^{(k)} \right) \ldots p \left( y_1^{(k)} | \boldsymbol{x}_1^{(k)} \right) \mu_0 \left( \boldsymbol{x}_1^{(k)} \right), \quad (2)$$

with $\mu_0 \left( \boldsymbol{x}_1^{(k)} \right)$ denoting an initial (action) distribution from which we sample the first decision $\boldsymbol{x}_1^{(k)}$. While it appears that off-the-shelf PPO can tackle the problem in Equation 1, Section 3.2 details difficulties associated with such an RL formulation, noting that in BO situations, the sparsity of reward signals can impede the learning of the network's weights $\boldsymbol{\theta}$. Before presenting those arguments, we now differentiate from the closest prior art, clarifying critical MDP differences.

**Connection & Differences to [13]** Although we are the first to propose end-to-end multi-task MDPs for meta-BO, others have also used RL to discover AFs, for example. Our parameterisation architecture and MDP definition significantly differ from the prior art, particularly from [13], the closest method to our work. Our approach uses a transformer-based deep network model to parameterise the whole pipeline (see Section 3.4). In contrast, the work in [13] assumes a pre-trained fixed Gaussian process surrogate and uses multi-layer perceptrons only to discover acquisitions. Our state variable requires historical information from which we jointly learn a probabilistic model and an acquisition end-to-end instead of requiring posterior means and variances of a Gaussian process model. Hence, our framework is more flexible, allowing us to model non-smooth black-box functions while overcoming some drawbacks of GP surrogates, like training and inference times.

## 3.2 Limitations of Regret Rewards in End-to-End Training

To define Equation 1, we followed the well-established literature of meta-BO [4, 13] and utilised simple regret reward functions. Although this choice is reasonable, we face challenges in applying such rewards in end-to-end training. Apart from difficulties associated with end-to-end training

of deep architectures [18], our RL algorithm is subject to additional complexities when estimating gradients from Equation 1 due to the sparsity of the reward function. To better understand this problem, we start by noticing that for a reward component $r_t^{(k)}$ to contribute to the cumulative summation $\sum_t \gamma^{t-1} r_t^{(k)}$, we need to observe a function value $y_t^{(k)}$ that outperforms all values we have seen so far, i.e., $y_t^{(k)} > \max_{1 \le \ell < t} y_\ell^{(k)}$. During the early training stages of RL, we can quantify the average number of such informative events (when $y_t^{(k)} > \max_{1 \le \ell < t} y_\ell^{(k)}$) by a combinatorial argument that frames this calculation as a calculation of the number of cycles in a permutation of $T^{(k)}$ elements, leading us to the following lemma.

**Lemma 3.1.** *Consider a task with a horizon length (budget) $T$, and define $r_t = \max_{1 \le \ell \le t} y_t$ the simple regret as introduced in Equation 1. For a history $\mathcal{H}_T$, let $m_{\mathcal{H}}$ denote the total number of informative rewards, i.e. the number of steps $t$ at which $y_t > \max_{1 \le \ell < t} y_\ell$. Under a random policy $\pi_\theta$, the number of informative events is logarithmic in $T$ such that: $\mathbb{E}_{\mathcal{H} \sim p_{\pi_\theta}}[m_{\mathcal{H}}] = \mathcal{O}(\log T)$, where $p_{\pi_\theta}$ is induced by $\pi_\theta$ as in Equation 2.*

We defer the proof of Lemma 3.1 to Appendix A due to space constraints. Here, we note that this result implies that the information contained in one sampled trajectory is sparse at the beginning of RL training when the policy acts randomly. Of course, this increases the difficulty of estimating informative gradients of Equation 1 when updating $\theta$. One can argue that the sparsity described in Lemma 3.1 only holds under random policies during the early stages of RL training and that sparsity patterns decrease as policies improve. Interestingly, simple regret rewards do not necessarily confirm this intuition. To realise this, consider the other end of the RL training spectrum in which policies have improved to near optimality such that $\pi_\theta \to \pi_{\theta^\star}$. Because $\pi_\theta$ has been trained to maximise regret, it will seek to suggest the optimal point of the current task, as early as possible in the BO trajectory. Consequently, the policy is encouraged to produce trajectories with *even sparser* rewards during later training stages, further complicating the problem of informative gradient estimates of Equation 1.

## 3.3   Inductive Biases and Auxiliary Tasks

Learning from sparse reward signals is a well-known difficulty in the reinforcement learning literature [32]. Many solutions, from imitation learning, [35] to exploration bonuses [36], improve reward signals to reduce agent-environment interactions and enhance gradient updates. Others [37] attempt to define more informative rewards from prior knowledge or via human interactions [38]. Unfortunately, both of those approaches are hard to use in BO. Indeed, manually engineering black-box-specific rewards is notoriously difficult and requires domain expertise and extensive knowledge of the source and target black-box functions we wish to optimise. Furthermore, learning from human feedback is data-intensive, conflicting with the goal of sample-efficient optimisation.

Another prominent direction demonstrating significant gains is the introduction of auxiliary tasks (losses) within RL that allows agents to discover relevant inductive biases leading to impressive empirical successes [19, 39, 40]. Inspired by those results, we propose introducing an inductive bias in our method via an auxiliary supervised loss. Since we have at our disposal the collected source tasks datasets on which we are training our architecture $\mathcal{D}^{(1)}, \ldots, \mathcal{D}^{(K)}$, we augment our objective such that our RL agent maximises not only rewards but also the likelihood of making correct predictions on these labelled datasets.

**Supervised Auxiliary Loss** Consider a source task $k$ and consider that we split its corresponding dataset into an observed set and a prediction set $\mathcal{D}^{(k)} = \mathcal{D}_{\text{obs}}^{(k)} \sqcup \mathcal{D}_{\text{perd}}^{(k)}$. We define the auxiliary loss to be exactly the log-likelihood of functions values $y^{(\text{pred})}$ at predicted locations $\boldsymbol{x}^{(\text{pred})}$ given observations $\mathcal{D}_{\text{obs}}$:

$$\mathcal{L}(\boldsymbol{\theta}) = \mathbb{E}_{k \sim p_{\text{task}}, \mathcal{D}_{\text{obs}}^{(k)}, \mathcal{D}_{\text{perd}}^{(k)}} \left[ \log p_{\boldsymbol{\theta}}(y_k^{(\text{pred})} | \boldsymbol{x}_k^{(\text{pred})}, \mathcal{D}_{\text{obs}}^{(k)}) \right]. \tag{3}$$

Interestingly, this part of our model specialises as a neural process (Section 2.2) with $\mathcal{D}_{\text{obs}}^{(k)}$ being the history $\mathcal{H}_t^{(k)}$ and $\boldsymbol{x}^{(\text{pred})}$ being the input points at which we wish to make predictions. To represent $p_{\boldsymbol{\theta}}(\cdot)$, we use a head of our transformer to predict multi-modal Riemannian (or bar plot probability density function) posteriors as [11, 41]. We compute Equation 3 on random iid splits of $\mathcal{D}^{(k)}$ and not directly on trajectories generated by the policy. This is because as the policy improves, the trajectories

it generates are composed of less diverse (non-iid.) points. Indeed as training progresses, $\pi_{\boldsymbol{\theta}}$ becomes better and therefore finds the optimal point in $\mathcal{D}^{(k)}$ more rapidly. To fully take advantage of the labelled data at our disposal, we evaluate this auxiliary loss on iid. data, i.e. splits of $\mathcal{D}^{(k)}$ into $\mathcal{D}^{(k)}_{\text{obs}}$ and $\mathcal{D}^{(k)}_{\text{pred}}$ sampled uniformly at random. It is sensible from a BO standpoint as well since we want the introduced inductive bias to encourage our network to maximise the likelihood not only in a neighbourhood of the optimiser but also in the rest of the dataset.

### 3.4 Neural Acquisition Processes

We now introduce in more detail our transformer architecture and note some of its important properties. We call this architecture the neural acquisition process (NAP) because it is a new type of NPs jointly predicting AF values and distribution over actions. Similarly to other NP [11, 12], it takes a context and queried locations as input. We parameterise it by $\boldsymbol{\theta}$ and denote the acquisition prediction by $\alpha_{\boldsymbol{\theta}}(\boldsymbol{x}^{(\text{pred})}, \mathcal{H}, t, T)$.

Since $\alpha_{\boldsymbol{\theta}}$ outputs real numbers, we still need to define how we can obtain a valid probability distribution over the action space to form a policy $\pi_{\boldsymbol{\theta}}(\cdot|\boldsymbol{s}_t)$. Defining such a distribution over the whole action space is hard if $\mathcal{A}$ is continuous. Hence, similarly to Volpp et al. [13], we evaluate the policy $\pi_{\boldsymbol{\theta}}$ only on the finite set of locations $\boldsymbol{x}^{(\text{pred})}$ for a given task[1]. Therefore, we have:

$$\pi_{\boldsymbol{\theta}}\left(\boldsymbol{x}^{(\text{pred})}_t|\boldsymbol{s}_t\right) \propto \frac{e^{\alpha_{\boldsymbol{\theta}}(\boldsymbol{x}^{(\text{pred})}_t, \mathcal{H}_t, t, T)}}{\sum_i^{n_{\text{pred}}} e^{\alpha_{\boldsymbol{\theta}}(\boldsymbol{x}^{(\text{pred})}_i, \mathcal{H}_t, t, T)}}.$$

We now have all the necessary components to define the full objective combining Equation 1 and 3: $\mathcal{J}(\boldsymbol{\theta}) = J(\pi_{\boldsymbol{\theta}}) + \lambda\mathcal{L}(\boldsymbol{\theta})$, where $\lambda$ is a hyperparameter to balance between the two objectives. We summarise the full training algorithm in Alg. 1 detailing each of the components.

Finally, we study some desirable properties of NAPs, explain how we can achieve them, and why they are important in the context of BO.

---

**Algorithm 1** Neural Acquisition Process training.

**Require:** Source tasks training data $\{\mathcal{D}^{(k)}\}_{k=1}^{K}$, initial parameters $\boldsymbol{\theta}$, budgets $T^{(k)} \equiv T$, discount factor $\gamma$, learning rate $\eta$

**for each** epoch **do**
    select task $k$ and dataset $\mathcal{D}^{(k)}$, set $\mathcal{H}_0 = \{\emptyset\}$
    **for** $t = 1, \dots, T$ **do**
      $x_t \sim \pi_{\boldsymbol{\theta}}(\cdot|\boldsymbol{s}_t)$      ▷ predict action
      $y_t = f^{(k)}(x_t)$      ▷ execute action
      $r_t = y^*_{\leq t}$       ▷ collect reward
      $\mathcal{H}_{t+1} \leftarrow \mathcal{H}_t \cup \{(x_t, y_t)\}$  ▷ update hist.
    **end for**
    $R = \sum_{t=1}^{T} \gamma^t r_t$      ▷ cumul. reward
    $\mathcal{D} \rightarrow \mathcal{D}_{\text{obs}} \sqcup \mathcal{D}_{\text{pred}}$     ▷ split source data
    $\mathcal{L} = p_{\boldsymbol{\theta}}(\boldsymbol{y}^{(\text{pred})}|\boldsymbol{x}^{(\text{pred})}, \mathcal{D}_{\text{obs}})$    ▷ aux. loss
    $\boldsymbol{\theta} \leftarrow \boldsymbol{\theta} + \eta(\nabla_{\boldsymbol{\theta}} R + \nabla_{\boldsymbol{\theta}} \mathcal{L})$   ▷ update $\boldsymbol{\theta}$
**end for**

---

**Property 3.2** (History-order invariance). *An NP $g$ is history-order invariant if for any choice of permutation function $\psi$ that changes the order of the points in history, $g(\boldsymbol{x}, \psi(\mathcal{H})) = g(\boldsymbol{x}, \mathcal{H})$.*

Unlike vanilla transformers, we do not use a positional encoding. It allows NAP to treat the history $\mathcal{H}$ as a set instead of an ordered sequence [42] and to be invariant to the order of the history. To form an $\langle \boldsymbol{x}, y \rangle$ pair in this set, we sum the embedding of $\boldsymbol{x}$ and $y$ [11], whereas for queried locations we only use the embedding of $\boldsymbol{x}$ for a token (see Fig. 1, left). It is important for BO since the order in which we collect the points is not relevant for assessing how promising is a new queried location. Previous meta-RL algorithms [43] also shown the importance of relying on order-invariance.

---

[1]With a continuous action space, one could optimise $\boldsymbol{x}^{(\text{pred})}$ to maximise the NAP outputs before forming the distribution over the discrete actions.

Table 1: We compare the properties of different transformer architectures. $L$ is the number of tokens needed to encode the meta-data, and $D$ denotes the dimensionality of $\mathcal{X}$.

| | History-order inv. | Query ind. | AF values | Tokens |
|---|---|---|---|---|
| NAP (ours) | ✔ | ✔ | ✔ | $t + n_{\text{pred}}$ |
| TNPs [12] | ✔ | ✘ | ✘ | $t + n_{\text{pred}}$ |
| OptFormer [41] | ✘ | ✘ | ✘ | $L + (D+2)(t + n_{\text{pred}})$ |
| PFN [11] | ✔ | ✔ | ✘ | $t + n_{\text{pred}}$ |

**Property 3.3** (Query independence). *A NP $g$ is query independent if for any choice of $n$ queried locations $\boldsymbol{x}^{(pred)} = (\boldsymbol{x}_1^{(pred)}, \ldots, \boldsymbol{x}_n^{(pred)})$, we have $g(\boldsymbol{x}^{(pred)}, \mathcal{H}) = (g(\boldsymbol{x}_1^{(pred)}, \mathcal{H}), \ldots, g(\boldsymbol{x}_n^{(pred)}, \mathcal{H}))$.*

In NAP, every token in the history can access each other through the self-attention mask, whereas the elements of $\boldsymbol{x}^{(pred)}$ can only attend to themselves and to tokens in the history $\mathcal{H}$ (see Fig. 1, right). Because the tokens inside $\boldsymbol{x}^{(pred)}$ cannot access each other and we do not use positional encoding, NAP is query independent, which is important to make consistent predictions of AF values for BO as they should not depend on the other queried locations. Additionally, NAP is fully differentiable enabling end-to-end training but also optimisation of the queried locations via gradient ascent for continuous action spaces. In Table 1, we highlight the differences with other state-of-the-art models regarding those properties.

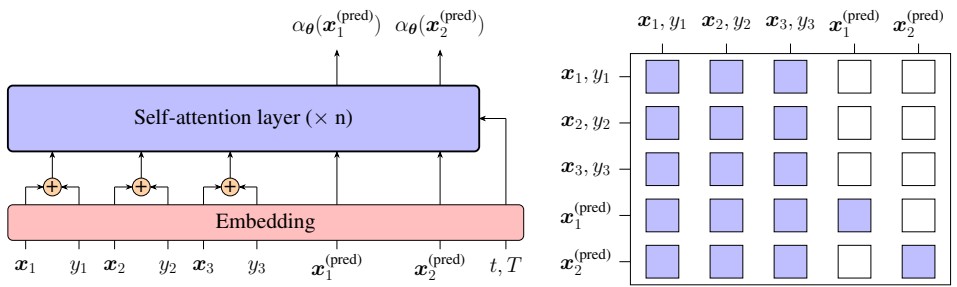

Figure 1: Our proposed NAP architecture (left) and an example of the masks applied during inference (right). We apply independent embedding on $\boldsymbol{x}_i$, $y_i$, $t$ and $T$. The colored squares mean that the tokens on the left can attend the tokens on the top in the self-attention layer.

## 4 Experiments

We conduct experiments on hyperparameter optimisation (HPO) and sequence optimisation tasks to assess NAP's efficiency. Regarding HPO, we run our algorithm on datasets from the HPO-B benchmark [44] and in real-world settings of tuning hyperparameters for Mixed-Integer Programming (MIP) solvers. For sequence optimisation, we test NAP on combinatorial black-box problems from antibody design and synthesis flow optimisation for electronic design and automation (EDA).

**Baselines** We compare our method against popular meta-learning baselines, including few-shot Bayesian optimisation (FSBO) [5], MetaBO [13] as well as OptFormers [41] when applicable. Moreover, we show how classical GP-based BO, equipped with an adequate kernel [2] and an EI acquisition function, which we title GP-EI, performs across domains. We first fit a GP on the meta-training datasets to enable a fair comparison. We initialise the GP model with those learnt kernel parameters at test time. This way, the GP baseline can benefit from the information provided in the source tasks. We also explore training a neural process directly on source task datasets and then using a fixed EI acquisition. For that, we introduce NP-EI that combines the same base architecture from [11] with EI. Additionally, we contrast NAP against random search (RS).

Following the standard practice in meta-BO, we report our results in terms of normalised regrets for easier comparison across all tasks. We attempt to re-implement all baselines across all empirical domains, extending previous implementations as needed, e.g., developing MetaBO [13] and FSBO [5] versions for combinatorial and mixed spaces to enable a fair comparison in MIP solver tuning, antibody and EDA design tasks.

**Remark on OptFormer** We re-run OptFormer with an EI AF on hyperparameter tuning tasks by extending the implementation in [41] to the discrete search space version of HPO-B. However, the lack of a complete open-source implementation and interpretable meta-data in the other benchmarks (e.g., in antibody design and EDA) prohibited successful execution.

---

[2]We adapt the GP kernel to each task depending on the nature of the optimisation variable, e.g., Matérn in continuous domains, categorical for combinatorial search space, and mixed when having both.

## 4.1 Hyperparameter Optimisation Results

**Hyperparameter Optimisation Benchmarks** We experiment on the HPO-B benchmark [44], which contains datasets of (classification) model hyperparameters and the models' corresponding accuracies across multiple types and search spaces. Due to resource constraints, we selected six representative search spaces. Nonetheless, we chose the search spaces to represent all underlying classification models in the experiment. We also always picked the ones with the least points to focus on the low data regime performance; see Appendix C.1 for more details. The results of our tests in Figure 2 demonstrate that NAP and OptFormer outperform all other baselines. Surprisingly, although NAP uses a much smaller architecture than OptFormer (around 15 million parameters vs 250 million for OptFormer) and trains on much less data (around 80k original points on average versus more than 3 million points for OptFormer), its regret performance is statistically similar to that of the OptFormer after 100 steps. Moreover, on the same GPU, to perform the same inference, NAP only uses 2% of OptFormer's compute time and around 40% of its memory usage.

**Tuning MIP Solvers** Apart from HPO-B, we consider another real-world example of hyperparameter tuning that requires finding the optimal parameters of MIP solvers. We use the open-source `SCIP` solver [45] and the `Benchmark` suite from the `MIPLib2017` [46] that consists of a collection of 240 problems. The objective is to find a set of hyperparameters so that `SCIP` can solve MIP instances in minimal time. Our high-dimensional search space comprises 135 hyperparameters with mixed types, including boolean, integers, categories and real numbers. We train our model on data collected from BO traces on 103 MIPs and test on a held-out set of 42 instances. Our results in Figure 2 demonstrate that NAP is capable of outperforming all other baselines reaching low regret about an order of magnitude faster than FSBO [5]. Figure 2 further demonstrates the importance of end-to-end training where NAP again outperforms NP-EI.

## 4.2 Sequence Optimisation Experiments

Now, we demonstrate NAP's abilities beyond hyperparameter tuning tasks in two real-world combinatorial black-box optimisation problems.

**Antibody CDRH3-Sequence Optimisation** This experiment focuses on finding antibodies that can bind to target antigens. Antigens are proteins, i.e., sequences of amino acids that fold into a 3D shape giving them specific chemical properties. A protein region called CDRH3 is decisive in the antibody's ability to bind to a target antigen. Following the work in [47], we represent CDRH3s as a string of 11 characters, each character being the code for a different amino acid in an alphabet of cardinality 22. The goal is to find the optimal CDRH3 that minimises the binding energy towards a specific antigen. Binding energies can be computed using state-of-the-art simulation software like Absolut! [48]. We collected datasets of CDRH3 sequences and their respective binding energies (with Absolut!) across various antigens from the protein database bank [49]. We then formed a transfer scenario across antigens where we meta-learn on 109 datasets, validate on 16, and test NAP on 32 new antigens. Our results in Figure 2 indicate that NAP is not limited to hyperparameter tuning tasks but can also outperform all other baselines in combinatorial domains.

**Electronic Design Automation (EDA)** Logic synthesis (LS) is an essential step in the EDA pipeline of the chip design process. At the beginning of LS, we represent the circuit as an AIG (an And-Inverter-Graph representation of Boolean functions) and seek to map it to a netlist of technology-dependent gates (e.g., 6-input logic gates in FPGA mapping). The goal in LS is to find a sequence of graph transformations such that the resulting netlist meets an objective that trades off the number of gates (area) and the size of the longest directed path (delay) in the netlist. We perform a sequence of logic synthesis operators dubbed a synthesis flow to optimise the AIG.

Following [50], we consider length 20 LS flows and allow an alphabet of 11 such operators, e.g., {`refactor`, `resub`, ..., `balance`} as implemented in the open-source `ABC` library [51]. We collected datasets for 43 different circuits. Each dataset consisted of 500 sequences (collected via a Genetic algorithm optimizer) and their associated area and delay. Additionally, we applied the well-known heuristic sequence `resyn2` on each circuit to get a reference area and delay. For this task, the black-box takes a sequence as input and returns the sum of area and delay ratios with respect to the reference ones, as detailed in Appendix B.1. We train all methods on 30 circuits from `OpenABC` [50], validate on 4 and test on 9. Our results in Figure 2 again demonstrate that NAP outperforms all other baselines by a significant margin.

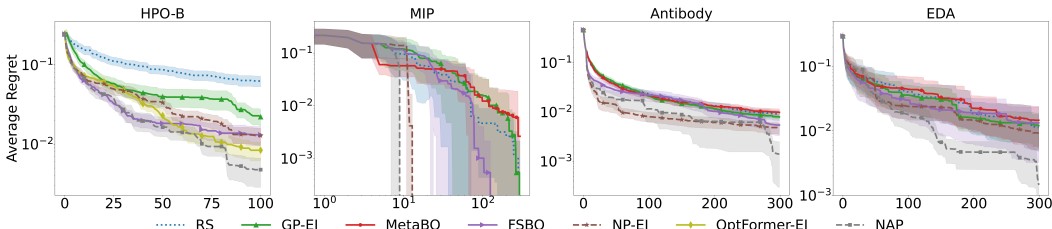

Figure 2: Average regret vs. BO iterations with 5 initial points. (Left) Results on 6 search spaces on the HPO-B benchmark. (Middle-left) Results tuning `SCIP` for solving 42 different MIPs. (Middle-right) Antibody CDR3 sequence optimisation on 32 test datasets corresponding to 32 different antigens. (Right) Logic synthesis operator sequence optimisation on 9 test datasets corresponding to 9 different circuits. For each method, error bars show confidence intervals computed across 5 runs on HPO-B and 10 runs on all the others.

# 5   Related Work

**Meta-Learning Paradigms**   Meta-learning aims to learn how to quickly solve a new task based on the experience gained from solving or observing data from related tasks [15]. To achieve this goal, we can follow two main directions. In the first one, the meta-learner uses source data to learn a good initialisation of its model parameters, such that it is able to make accurate predictions after only few optimisation steps on the new target task [52]. The second one is more ambitious as it endeavours to directly learn, from the related tasks, a general rule to predict a helpful quantity to suggest the next point (black-box value, acquisition function value, etc.) from a context of observations (i.e. our history $\mathcal{H}^{(k)}$). Works following that direction, which include ours, can rely on different types of models to learn this general rule, such as NPs [10], recurrent neural networks [53], HyperNetworks [16] or transformers [12, 11]. Orthogonal to these two directions is the learning of a hyper-posterior on the datasets to meta-train on. In that line of research, Rothfuss et al. [54] suggest the use of Stein Variational Gradient Descent (SVGD) to estimate uncertainty with multiple models better, and Hsieh et al. [14] extend the use of SVGD to meta-learn AFs. We consider those last two works as an orthogonal direction to our, as NAP could also benefit from SVGD updates.

**Learning a Meta-Model**   Chen et al. [55] and TV et al. [56] train an RNN to predict what should be the next suggestion in BO instead of predicting the acquisition function value. We do not compare to their method given the lack of available implementation. Still, we compare to the outperforming approach developed by Chen et al. [41] based on a transformer architecture designed to do meta-BO on hyperparameter tuning problems. Their OptFormer is trained on related tasks over different search spaces to output the next point to evaluate and predict its value. The next point is sequentially decoded, one token at a time, and exploits the hyperparameters' names or descriptions to improve generalisation across tasks. Contrary to NAP, OptFormer is not designed to predict acquisition value at any point and does not meet the two properties 3.2 and 3.3, and therefore needs much more training data to predict the proper sequence of tokens. We note that NAP does not rely on variable descriptions, making it easily deployable on various tasks and still very competitive in the hyperparameter optimisation context.

Rather than learning an entire predictive model from scratch, prior works [5, 7–9] learn deep kernel surrogates, i.e. a warping functions mapping the input space with a neural network before it is given to a GP. Learning only the input transformation allows the authors to rely on the closed-form posterior prediction capacity of standard GP models. To perform the transfer, Feurer et al. [6] rely on an ensemble of GPs. Iwata [57] is an approach close to FSBO [5] as it learns a meta-model by meta-training a Deep Kernel GP. Notably, it does so in an end-to-end fashion using RL to propagate gradients through the acquisition function and the GP back to the deep kernel. It does not, however, learn a meta-acquisition.

**Learning a Meta-Acquisition Function**   Hsieh et al. [14] and Volpp et al. [13] choose to perform transfer through the acquisition function. They use directly GP surrogates and define the acquisition as a neural network that is meta-trained on related source tasks. They first pre-train GP surrogates on all source tasks and fix their kernel parameters. They then rely on RL training to meta-learn the

neural acquisition function that takes as inputs the posterior mean and variance of the GP surrogate (which is itself trained online at test time). At test time, they allow for update in the GP but keep the weights of the neural acquisition fixed.

In summary, the methods meta-learning AFs do not do so in an end-to-end fashion and still rely on trained GP surrogates. While these methods are principled and competitive, they suffer from the cost of inverting the GP kernel matrix (cubic in the number of observations). In comparison, NAP can make predictions through a simple forward pass. On the other hand, the methods that learn a meta-model, either use a Deep Kernel GP (suffering the same cost) or have to learn a large model from scratch, costing a lot of budgets for collecting data beforehand, as well a pre-training time. Both of them use standard acquisition functions, missing the potential benefits of doing transfer in acquisition between tasks.

The performance of some of those algorithms on HPO-B is presented in Appendix-C showing that despite learning an architecture from scratch, NAP achieves a lower regret. For a more detailed survey on transfer and meta-learning in BO, we refer the reader to Bai et al. [4].

## 6   Conclusion

We proposed the first end-to-end training protocol to meta-learn acquisition functions in meta-BO. Our method predicts acquisition values directly from a history of points with meta-reinforcement learning and auxiliary losses. We demonstrated new state-of-the-art results compared to popular baselines in a wide range of benchmarks.

**Limitations & Future Work** Our architecture suffers from the usual quadratic complexity of the transformer in terms of the number of tokens which limits the budget of BO steps. Nevertheless, it can still handle around 5000 steps, which is enough for most BO scenarios. Another limitation of our architecture is that we need to train a new model for each search space. In future, we plan to enable our method to leverage meta-training from multiple search spaces and investigate how we could design data-driven BO-specific augmentation to further mitigate meta-overfitting [58].

## Acknowledgements

We would like to thank Prof. Frank Hutter and his team for their constructive feedback and for the availability of their benchmark code and datasets. We also would like to thanks Massimiliano Patacchiola for his comments during the writing phase of the paper.

This work is supported by the CSTT on Generalisable Robot Learning via Machine Learning Models 2100332-GB and by the 2030 "New Generation of AI" - Major Project of China under grant No. 2022ZD0116408.

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
