# OpenReview forum: "End-to-End Meta-Bayesian Optimisation with Transformer Neural Processes"
_NeurIPS.cc/2023/Conference — NeurIPS 2023 poster_

### Official Review · Reviewer_dCjX · 2023-07-03

**Soundness:** 4 excellent
**Presentation:** 4 excellent
**Contribution:** 3 good
**Rating:** 7
**Confidence:** 4

**Summary:**

The authors propose Neural Acquisition Process, a novel method for Bayesian optimization by meta-learning a function that jointly performs the surrogate and acquisition steps. The function, i.e. a transformer, is able to work as a policy that predicts the action **a** given a state **s**, where the action is the hyperparameter to observe. The state comprises the history of the observed samples. Also, the transformer can predict probabilistic the performance of a hyperparameter. They use reinforcement learning to train the policy and, additionally, they use negative log-likelihood of the predicted performance for meta-tasks as an auxiliary loss. The authors test the method on 4 different benchmarks and delineate its difference from previous work, demonstrating the relevance of their approach.

**Strengths:**

* They present a novel and effective method for pre-training a transformer to perform Bayesian optimization. The method is well-founded and all the parts make sense from an engineering perspective. For instance, the use of auxiliary losses and reinforcement learning appears as a very important and interesting approach.
* They establish a difference from prior work in terms of some parameters such as history order invariance, F values, number of tokens, etc.
* The authors provide a strong experimental protocol, comparing to relevant baselines in a broad set of datasets and benchmarks, which makes the empirical results very strong.


**Weaknesses:**

* The method might be overkill for small search spaces and nonexpensive functions. However, many black-box functions nowadays, such as deep neural networks, are expensive to evaluate, thus this approach might be highly relevant.
* The authors do not present a comparison against time. How does the method perform in terms of time?

**Questions:**

* Why is there no confidence bar for the MIP dataset?

**Limitations:**

No important limitation is detected.

---

> ### Author Rebuttal · Authors · 2023-08-09
>
> We thank the reviewer for their time and their remarks as well as underlining the soundness of our experimental protocol. We try to reply to their concerns below.
>
> >The method might be overkill for small search spaces and nonexpensive functions. However, many black-box functions nowadays, such as deep neural networks, are expensive to evaluate, thus this approach might be highly relevant.
>
> Indeed, for small search spaces and non-expensive black-box functions, our method might be overkill, and so would regular BO to a certain extent because if querying the objective is cheap, sample efficiency becomes less important than time efficiency. If the objective becomes very expensive to evaluate, then black-box solvers become useful. We keep our experiments limited to this case, in which sample efficiency is most important as the main bottleneck in terms of cost and time is querying the objective, and this cost outweighs the cost of any model fitting or pretraining done offline or online. We conduct experiments in search spaces that can be considered small (e,g, in HPO-B) but also in larger ones, e.g. Antibody design and EDA in combinatorial space and MIP in high-dimension and mixed space (up to 135 dimensions).
>
> >The authors do not present a comparison against time. How does the method perform in terms of time?
>
> We have to distinguish the time during testing and the time during pretraining. Methods that make use of GP surrogates are bound to be slower in terms of time compared to NAP and NP-EI that only use forward passes in the network to decide what is the next query point. At each step of BO at test time, for example FSBO [1] or MetaBO [2] have to fit a GP model. However, pretraining time is more important for NAP, and MetaBO compared with NP-EI as the latter doesn't use RL which is computationally more costly during offline pretraining. They also take more time to pretrain than FSBO that meta-learns a feature extractor for a GP (deep kernel). The GP-EI baseline has no pretraining as it is a classical BO method with no amount of meta-learning. Finally, we mention a more detailed comparison with Optformer-EI [3] in appendix B.2. showing that NAP is faster to train, evaluate and uses far less memory.
>
> It is natural that there is a tradeoff between pretraining cost and regret performance at test time, but as the most important metric in our setting is sample efficiency at test time, we did not show the results versus time.
> During testing, we observed a rough estimate of 10x speedup compared to GPs-based methods that requires refitting. We will add those numbers to the appendix for reference.
>
> >Question: Why is there no confidence bar for the MIP dataset?
>
> The error bars are visible if you open the PDF file in Acrobat Reader. We also found that they are visible when opening it with Firefox. But indeed, we are not sure why they are sadly not visible when opening the PDF with Chrome. We are working on finding where this issue comes from.
>
> ----------
>
> [1] Martin Wistuba and Josif Grabocka. Few-shot bayesian optimization with deep kernel surrogates. In 9th International Conference on Learning Representations, ICLR 2021, Virtual Event, Austria, May 3-7, 2021.
>
> [2] Michael Volpp, Lukas P. Fröhlich, Kirsten Fischer, Andreas Doerr, Stefan Falkner, Frank Hutter, and Christian Daniel. Meta-learning acquisition functions for transfer learning in bayesian optimization. In 8th International Conference on Learning Representations, ICLR 2020, Addis Ababa, Ethiopia, April 26-30, 2020.
>
> [3] Yutian Chen, Xingyou Song, Chansoo Lee, Zi Wang, Richard Zhang, David Dohan, Kazuya Kawakami, Greg Kochanski, Arnaud Doucet, Marc Aurelio Ranzato, Sagi Perel, and Nando de Freitas. Towards learning universal hyperparameter optimizers with transformers. In S. Koyejo, S. Mohamed, A. Agarwal, D. Belgrave, K. Cho, and A. Oh, editors, Advances in Neural Information Processing Systems, volume 35, pages 32053–32068. Curran Associates, Inc., 2022.

---

> > ### Comment · Reviewer_6XK8 · 2023-08-14
> > **Clarification**
> >
> > The authors may excuse my unclear comment with respect to time. What I was interested in was test time and costly it is. For example, if you consider the compute overhead of your smart searcher into account, is a different able to evaluate more configurations in the same time and therefore find a better solution? Effectively Figure 2 with wall clock time on the x-axis.
> >
> > Edit: Sorry, not my review. I'm interested anyway :D

---

> > > ### Author Response · Authors · 2023-08-15
> > >
> > > As a followup to our answer to reviewer dCjX as well as this official comment from reviewer 6XK8, we summarise some average test running time results in the hope to provide both with more details. These results will be concatenated in a table that we are happy to add to the appendix.
> > >
> > > As mentioned before in our rebuttal of reviewer dCjX, we are in a BO setting where querying the black-box is considered to be the main bottleneck. For example, in the antibody experiment, this could be true both in terms of monetary and time costs as evaluating the objective could mean manufacturing the molecule and testing it in a wet-lab experiment. Furthermore, we do not have the actual times of the black-box evaluations for some experiments. For instance, the authors of the HPO-B dataset do not report those numbers. These are rather costly models to train and test and it would be prohibitively expensive to query the black-box ourselves. We use result files posted on the authors' repository which only contain black-box values for some baselines. Similarly, we do not have the real black-box evaluation time for the Antibody experiment as the data collection was done through a simulator.
> > >
> > > What we do have, however, is the time to evaluate the black-box in the MIP and EDA experiments. By design of the experiment, evaluating one set of hyperparameters on the MIP experiment takes 2 hours. Compared to that, the time to train a GP model or doing a forward pass in NAP at test time is negligible. On EDA, the black-box time depends on the circuit so we approximate an average running time of 1 minute per circuit on open-source circuits, but this can take several hours on industrial circuits.
> > >
> > > The tables below compares the average test time of 1 seed across various methods (from figure 2 in the paper) without the black-box time taken into account, then with the black-box time taken into account and finally with both the black-box time and the pretraining time taken into account.
> > >
> > > In the first column, we can see that methods which have to fit a GP during the BO loop (FSBO, MetaBO and GP-EI) are considerably slowed down compared to methods like NAP that only do forward passes through their network. This is because fitting the GP surrogate at each BO step is time consuming, and increasingly so, as its dominant computational cost is cubic in the number of observed points. Note also that FSBO not only fits a GP at each step but also fine tunes the MLP of its deep kernel, hence the extra time.
> > >
> > > The second column with the black-box time taken into account further underlines that even though NAP is faster at test time than e.g. FSBO or GP-EI, this time gain it is negligible compared to the blackbox evalutions.
> > >
> > > The third column takes into account the pretraining time for methods that require it. Note that for different test functions within the same search space, we can reuse the same model for NAP, NP-EI, MetaBO and FSBO without having to redo the pretraining, so we divided the pretraining time by the number of seeds and test functions.
> > > Hence, it does not add much time to the total.
> > > It should be underlined that this way of presenting BO results is less readable than presenting regret vs BO steps as the more seeds and test tasks we have, the more negligible the pretraining time becomes compared to the black-box evaluation time.
> > >
> > > **Table 1: MIP experiment - average test time of 1 seed**
> > > | Method | without bbox | with bbox | with bbox & pretrain |
> > > | --------------- | --------------- | --------------- | --------------- |
> > > | GP-EI | 585sec | 25d 0hr 9min 45sec | 25d 0hr 9min 45sec |
> > > | FSBO | 330sec | 25d 0hr 5min 30sec | 25d 0hr 10min |
> > > | MetaBO | 30sec | 25d 0hr 0min 30sec | 25d 0hr 12min |
> > > | NP-EI | 2sec | 25d 0hr 0min 2sec | 25d 0hr 36min |
> > > | NAP | 3sec | 25d 0hr 0min 3sec | 25d 1hr |
> > >
> > > **Table 2: EDA experiment - average test time of one seed**
> > > | Method | without bbox | with bbox | with bbox & pretrain |
> > > | --------------- | --------------- | --------------- | --------------- |
> > > | GP-EI | 17sec | 1hr 5min 17sec | 1hr 5min 17sec |
> > > | FSBO | 516sec | 1hr 13min 36sec | 1hr 14min 6sec |
> > > | MetaBO | 35sec | 1hr 5min 35sec | 1hr 12min 5sec |
> > > | NP-EI | 8sec | 1hr 5min 8sec | 1hr 7min 34sec |
> > > | NAP | 9sec | 1hr 5min 9sec | 1hr 7min 42sec |

---

> > > > ### Comment · Reviewer_dCjX · 2023-08-17
> > > > **Reply to authors**
> > > >
> > > > Thanks for the time and effort put into the rebuttal. I consider raising my score from 6 to 7.

---

### Official Review · Reviewer_Mquj · 2023-07-05

**Soundness:** 3 good
**Presentation:** 3 good
**Contribution:** 3 good
**Rating:** 6
**Confidence:** 3

**Summary:**

This paper proposed an end-to-end transformer-based framework for meta-Bayesian optimization (meta-BO). It formulated the meta-BO as a RL problem, defining a MDP in which a policy can be trained to solve the meta-BO problem. To help the training of RL algorithm, this paper also proposed an inductive bias via an auxiliary loss. The experiments had demonstrated the effectiveness of the proposed method.

**Strengths:**

1. This paper deals with a good problem, which has a lot of applications in real world. The formulation of MDP is also reasonable.
2. This paper is well written, and also easy to follow.
3. The proposed inductive bias via an auxiliary loss Section 3.3. is insightful.


**Weaknesses:**

1. The experiments is insufficient. In the current experiments, the author only provided the evaluation results of the proposed method NAP and other baselines on several benchmarks. However, it would be better to do some ablation studies to further analyze NAP. For example, it should be important to investigate the effect of the auxiliary loss Eq(3).
2. The applications are limited. RL is known to be sample inefficient, especially when the state/action spaces are of high dimensionality. Furthermore, NAP utilized a transformer to extract historic information, which is also relatively hard to train. Therefore, NAP may be implicitly restricted into situations when x is of low dimension, limiting its applications.


**Questions:**

Please refer to weakness.

**Limitations:**

Yes.

---

> ### Author Rebuttal · Authors · 2023-08-09
>
> We thank the reviewer for their positive comments on the applicability of our work and we try to answer their concerns below.
>
> >The experiments is insufficient. In the current experiments, the author only provided the evaluation results of the proposed method NAP and other baselines on several benchmarks. However, it would be better to do some ablation studies to further analyze NAP. For example, it should be important to investigate the effect of the auxiliary loss Eq(3).
>
> We would like to highlight that we did an ablation study in Section C.1 of the Appendix.
> In Table 3 we show for each version of NAP, the components that are present or absent. Figure 4 shows the regret results on another experiment, an HPO experiment on XGBoost hyperparameters. We show that NAP-RL, trained only through the RL loss, performs worse than NAP, emphasising the importance of the auxiliary loss for downstream performance.
>
> >The applications are limited. RL is known to be sample inefficient, especially when the state/action spaces are of high dimensionality. Furthermore, NAP utilized a transformer to extract historic information, which is also relatively hard to train. Therefore, NAP may be implicitly restricted into situations when x is of low dimension, limiting its applications.
>
> Note that we pre-train NAP with RL, as such it is **offline** and therefore does not impact the **online** sample efficiency of NAP at test time as we only use online observations to select the next query.
> In BO in general, the metric that matters most is online sample efficiency because we assume that we are in a setting where the black-box objective is costly and evaluating it is the main bottleneck.
>
> Furthermore, we show that even with a dataset of rather limited size, we can pre-train NAP e.g. on the MIP experiment where the search space is of high dimension (135), and are still able to learn useful information transferable to new tasks that leads to best performance. We do acknowledge some limitations, for instance that NAP is restricted to be trained and tested on a search space of similar dimension (discussed in Limitations section) and we would like to extend it to be dimension-agnostic in a future work. Finally, we acknowledge the difficulty of training a transformer with RL signals only (underlined in the ablation), hence our use of the auxiliary loss.

---

> > ### Comment · Reviewer_Mquj · 2023-08-18
> >
> > Thanks for your responses.

---

### Official Review · Reviewer_6XK8 · 2023-07-05

**Soundness:** 3 good
**Presentation:** 3 good
**Contribution:** 3 good
**Rating:** 5
**Confidence:** 5

**Summary:**

The authors present a hyperparameter optimization method that is based on a transformer. In contrast to other recent work on HPO with transformers, this method does not rely on an acquisition function but instead trains an end-to-end model that outputs the acquisition scores directly.

**Strengths:**

**Diversity of benchmark tasks**

Experiments on 4 different types of tasks, comparison against state-of-the-art methods. Demonstrate some improvements over baselines.

**Clear Motivation**

Use of transformers have shown promising results. End-to-end learning is always a good idea. The authors combine these previously existing ideas.

**Weaknesses:**

**Complexity of the Method**

The authors combine transformers, reinforcement learning and neural processes. There are most likely a lot of knobs and using this method effectively might be very difficult. Meta-BO is a prime example of a method that adds quite a bit of complexity and it turned out that it did not generalize to any other setup than considered by the authors.

**Novelty**

This is neither the first paper to use transfer learning, transformers, RL, or end-to-end learning for HPO. This is a paper that combines all of it to a new method.
A paper not discussed in this work: Bing-Jing Hsieh, Ping-Chun Hsieh, Xi Liu: Reinforced Few-Shot Acquisition Function Learning for Bayesian Optimization. NeurIPS 2021: 7718-7731
It also trains an end-to-end model and the authors demonstrated some improvements over FSBO.

**Some Overclaiming of Contributions**

In multiple locations, the authors claim to present the " first end-to-end training protocol for meta-BO". This is not the case, see reference above and MetaBO.

**Questions:**

This is a complex method that probably introduce several hyperparameters itself. How do you set them? How robust are they? Do you think there is a chance that this method can be directly applied to a different task or will it require significant work to get it running?

Why is NP-EI missing for HPO-B?

**Limitations:**

The authors do not address the limitations that come with the complexity of this method, i.e., additional hyperparameters.

---

> ### Author Rebuttal · Authors · 2023-08-09
>
> We thank the reviewer for their time and for stressing the importance of end-to-end training. We try to answer the questions regarding complexity, novelty and hyperparameter tuning below.
>
> >The authors combine transformers, reinforcement learning and neural processes. There are most likely a lot of knobs and using this method effectively might be very difficult. Meta-BO is a prime example of a method that adds quite a bit of complexity and it turned out that it did not generalize to any other setup than considered by the authors.
>
> >This is a complex method that probably introduce several hyperparameters itself. How do you set them? How robust are they? Do you think there is a chance that this method can be directly applied to a different task or will it require significant work to get it running?
>
> As this comment and question relate to similar topics, we answer them together. Our method involves a good amount of pretraining on source task data but this is not unlike other meta-BO methods. In our experiments we give the same budget for pretraining to NAP and MetaBO. The hyperparameters used are all provided in Table 2 in appendix and they remain the same across our experiments.  We set their values based on the ones found in the repository of the MetaBO [1] baseline and did not change them further. We also fix an equal weight between the RL loss and the supervised auxiliary loss. To preserve fair comparison between all methods and because we assume the hyperparameters set by the authors of each baseline are already optimised, we do not tune ours. In that regard, we believe that this makes NAP more portable to new tasks as the hyperparameters have not been tuned for each specific task.
>
> We acknowledge the difficulty of applying MetaBO in other domains. However, we believe that it comes from hypotheses made by the authors (that the parameters of the GPs are fixed during training and testing).
> By tackling 9 heterogeneous tasks (6 in HPO-B), we show that NAP is not suffering from the same problem and achieve good performance across tasks.
>
> >Novelty: This is neither the first paper to use transfer learning, transformers, RL, or end-to-end learning for HPO. This is a paper that combines all of it to a new method. A paper not discussed in this work: Bing-Jing Hsieh, Ping-Chun Hsieh, Xi Liu: Reinforced Few-Shot Acquisition Function Learning for Bayesian Optimization. NeurIPS 2021: 7718-7731 It also trains an end-to-end model and the authors demonstrated some improvements over FSBO.
>
> >Some Overclaiming of Contributions: In multiple locations, the authors claim to present the " first end-to-end training protocol for meta-BO". This is not the case, see reference above and MetaBO.
>
> These two remarks are also linked so we answer them together. We refer to the work by Hsieh et al. on a few occasions in our paper (ref [14] in bib, [2] in this comment) but thank you for giving us a chance to expand on the differences between their approach and ours. We will also add these arguments in the Related Work section.
>
> First, we would like to outline that the setting in [2] is slightly different from our setting in that it aims to do *adaptation*, i.e. few-shot learning; which means that the authors meta-learn a neural acquisition on some source task data as well as data generated from a prior, but then allow their model to have access to test task observations to fine tune. Moreover, as PACOH [3], they rely on ensemble-method with Stein Variational Gradient Descent (SVGD). We consider these prior works as orthogonal directions as they can be combined with NAP: we could also use SVGD with multiple transformers or to finetune NAP at test time.
>
> Second, we can argue that FSAF is not end-to-end differentiable because it still relies on Gaussian Process surrogates (also the case for the MetaBO [1] baseline) and hence is classified as a two stage approach. In their paper in section 3.1 the authors rely on the posterior mean and variance of a GP for state-action representation of each point.
>
> What we are claiming is to be the first **end-to-end differentiable** method for meta-BO that specifically generalises neural processes to learn acquisition functions directly from the raw observations. We will make that distinction clearer in the paper, in abstract and introduction when talking about contributions.
>
> >NP-EI missing for HPO-B
>
> Thank you for pointing that out, we added NP-EI as a baseline in the HPO-B experiment, please see figures of the uploaded rebuttal PDF.
>
> ----------
>
> [1] Michael Volpp, Lukas P. Fröhlich, Kirsten Fischer, Andreas Doerr, Stefan Falkner, Frank Hutter, and Christian Daniel. Meta-learning acquisition functions for transfer learning in bayesian optimization. In 8th International Conference on Learning Representations, ICLR 2020, Addis Ababa, Ethiopia, April 26-30, 2020.
>
> [2] Bing-Jing Hsieh, Ping-Chun Hsieh, and Xi Liu. Reinforced few-shot acquisition function learning for bayesian optimization. In M. Ranzato, A. Beygelzimer, Y. Dauphin, P.S. Liang, and J. Wortman Vaughan, editors, Advances in Neural Information Processing Systems, volume 34, pages 7718–7731. Curran Associates, Inc., 2021.
>
> [3] Jonas Rothfuss, Vincent Fortuin, Martin Josifoski, and Andreas Krause. PACOH: bayes-optimal meta-learning with pac-guarantees. In Marina Meila and Tong Zhang, editors, Proceedings of the 38th International Conference on Machine Learning, ICML 2021, 18-24 July 2021, Virtual Event, volume 139 of Proceedings of Machine Learning Research, pages 9116–9126. PMLR, 2021.

---

> > ### Comment · Reviewer_6XK8 · 2023-08-15
> > **Follow-up**
> >
> > > FSAF is not end-to-end differentiable because it still relies on Gaussian Process surrogates
> >
> > Can you elaborate in what way GP surrogates make it non-differentiable?

---

> > > ### Author Response · Authors · 2023-08-15
> > >
> > > We apologise if our answer was misleading, what we meant was that FSAF (Hshieh et al., 2021) and MetaBO (Volpp et al., 2020) are not end-to-end differentiable because the authors use **pretrained** GP surrogates. Indeed, during the pretraining of the neural AF, the authors use a GP surrogate with parameters fit on data from the source task. They then keep those parameters fixed during RL training of the neural AF. Because of that, their approach is considered to be a two-stage approach.
> > >
> > > In theory, however, it would be possible to train a GP surrogate as well as a neural acquisition function in an end-to-end fashion, through RL. We see at least two options for such a setup.
> > >
> > > The first option would be to pretrain a GP on all source task data and then, during RL training of the neural AF, also take gradient steps in the GP to update its mean, kernel and likelihood parameters.
> > >
> > > The second option would be to directly train a GP from scratch together with the neural AF with RL and an auxiliary loss that would correspond to the marginal likelihood.
> > >
> > > This would be an entirely new baseline as we are not aware of any work that is training a GP surrogate and a neural acquisition function completely end-to-end in a way that corresponds to the options described above.
> > > If the reviewer still would like this to be studied, we are open to setting up new experiments to test this method, however, we are not sure how well that would work in practice and we have limited time at our disposal until the end of this discussion.

---

### Official Review · Reviewer_K11E · 2023-07-07

**Soundness:** 3 good
**Presentation:** 3 good
**Contribution:** 3 good
**Rating:** 6
**Confidence:** 3

**Summary:**

This work proposes an approach to meta-Bayesian optimization (BO) that features a single transformer neural process architecture for both the meta-surrogate function and meta-acquisition function, which are respectively trained via standard neural process maximum likelihood and model-free meta-reinforcement learning (in the style of RL^2) frameworks. The authors demonstrate the empirical benefits of such an approach over prior meta-BO methods on four suites of black-box optimization problems.

**Strengths:**

### Originality
My background is in meta-learning and reinforcement learning, not Bayesian optimization. It appears that the proposed approach is novel for meta-BO in that it makes minimal modelling choices, opting for a completely black-box neural process for the meta-surrogate function as opposed to e.g. a Gaussian process, and a completely black-box acquisition function as opposed to e.g. expected improvement (EI).

### Quality
The empirical evaluation seems reasonably thorough (though again, I am not an expert in BO), with experiments based on multiple data sources and a healthy pool of competing methods. I also appreciated the described engineering effort in adapting previous methods to handle datasets for which they were not originally designed for.

### Clarity
I'm not sure that Lemma 3.1 adds much substance in favor of the paper's design decisions. Instead, perhaps we can more loosely but intuitively argue that typical acquisition functions such as EI depend on the surrogate function's predictions of the underlying objective function (as it well should, to avoid no free lunch), but this dependency is dropped with a completely end-to-end black box acquisition function. The neural process meta-surrogate training can then be justified as softly reintroducing this dependency a la multi-task learning (multi in the sense of surrogate function modeling and acquisition function policy learning, not multiple BO tasks) via a shared architecture.

### Significance
I imagine the demonstrated empirical benefit of an end-to-end solution in a domain where handcrafted solution components have been standard will have considerable significance.

**Weaknesses:**

- A minor point -- it is unusual to see one maximizing a "loss" (Eq. 3 and Alg. 1).

**Questions:**

- How is your framing of the acquisition function as a policy with order-invariant processing of the current optimization trace related to ideas in context-based/prototype-based meta-reinforcement learning [A]? It seems like the former is a specific instance of the latter where the state space for each MDP is singleton.

[A] Rakelly & Zhou et al., Efficient Off-Policy Meta-Reinforcement Learning via Probabilistic Context Variables, ICML 2019.

**Limitations:**

Yes.

---

> ### Author Rebuttal · Authors · 2023-08-09
>
> We thank the reviewer for their comment, for the remark on extending baselines to search spaces they were not designed for originally, and for raising an interesting point regarding meta-RL, We will answer point by point below.
>
> >I'm not sure that Lemma 3.1 adds much substance in favor of the paper's design decisions. Instead, perhaps we can more loosely but intuitively argue that typical acquisition functions such as EI depend on the surrogate function's predictions of the underlying objective function (as it well should, to avoid no free lunch), but this dependency is dropped with a completely end-to-end black box acquisition function. The neural process meta-surrogate training can then be justified as softly reintroducing this dependency a la multi-task learning (multi in the sense of surrogate function modeling and acquisition function policy learning, not multiple BO tasks) via a shared architecture.
>
> Regarding clarity, the lemma 3.1 is a formal explanation of the sparsity of the reward used in RL, specifically linked to the use for BO.
> This issue is well-known in many RL problems but we wanted to make it clear that it is also the case in a meta-BO setting. This in itself, does justify the use of an auxiliary loss, but we also agree with the intuition you mentioned for the choice of the chosen auxiliary loss.
>
> >How is your framing of the acquisition function as a policy with order-invariant processing of the current optimization trace related to ideas in context-based/prototype-based meta-reinforcement learning [A]? It seems like the former is a specific instance of the latter where the state space for each MDP is singleton.
>
> If we want to match the history order-invariance of our policy with the tuple order-invariance of the mentioned paper, the tuple $t_0$, ..., $t_{n-1}$ without the reward should be defined like $t_0 = (s=\{\}, a=x_0, s'=f(x_0))$ and $t_i = (s=x_{i-1}, a=x_i, s'=f(x_i))$.
> We believe this is what you meant by a singleton state-space.
> If it is doable in practice, it does not form a standard MDP because the reward function cannot be defined based on a single tuple.
>
> Instead, a better instantiation would define the state in the tuple as a history of collected points $(x_i, f(x_i))$, however, in such case, the history order-invariance is not necessarily preserved through tuple order-invariance.
>
> This is an interesting point however, and we are going to include the mentioned paper in the related work section.

---

> > ### Comment · Reviewer_K11E · 2023-08-14
> > **Confidence Update**
> >
> > Thank you for your responses to my questions. I have increased my rating confidence to a 3.

---

### Official Review · Reviewer_P5Ji · 2023-07-09

**Soundness:** 3 good
**Presentation:** 4 excellent
**Contribution:** 2 fair
**Rating:** 5
**Confidence:** 2

**Summary:**

In this work, they developed the first end-end transformer based architecture for meta Bayesian Optimization and demonstrated empirically state-of-the-art regret minimization on hyperparameter optimization, antibody and chip design problems. They propose using a novel transformer architecture based Neural Process to learn the acquisition function and train it E2E compared to prior work in meta-learning acquisition functions. They identified logarithmic reward sparsity patterns in RL and introduced an auxiliary loss maximizing log-likelyhood of making the correct predictions on their labeled source task datasets as an inductive bias to stabilize training.

**Strengths:**

Demonstrates strong empirical results on their tasks and ablation experiments are included in the appendix demonstrating the importance of their inclusions.

Proposes novel combination of transformer network and E2E learning of the acquisition function for Bayesian optimization with a new formulation and inclusion of auxiliary loss.

Well written and well positioned with existing work.

**Weaknesses:**

Somewhat limited novelty due to prior separately have utilized transformers for Bayesian optimization and other prior work having meta-learned acquisition functions.

Somewhat limited experimental evidence since no hyperparameter search was conducted on the proposed method or baseline. Results were only included on four tasks.

**Questions:**

Did you tune the weighting of the auxiliary loss?

How were the hyperparameters chosen for the implementation if no hyperparameter search was conducted.

**Limitations:**

Discussed limitations limiting to 5000 BO steps due to their use of the transformer network.

---

> ### Author Rebuttal · Authors · 2023-08-09
>
> We thank the reviewer for their positive comments and for underlining the clarity of the writing. We thank them for raising interesting points that we will try to explain below.
>
> > Somewhat limited novelty due to prior separately have utilized transformers for Bayesian optimization and other prior work having meta-learned acquisition functions.
>
> As discussed in related work, prior work indeed exists where authors use RL to meta-learn neural acquisition functions [1,2]. However they rely on Gaussian Process models and by consequence their method is not end-to-end trainable.  Prior work also exists where authors use transformers but their goal is to meta-learn a surrogate model [3,4] which can in turn be used along with an off-the-shelf acquisition function. Again, these methods do not learn a single architecture that directly predicts acquisition function values from a history of observations. One of our contributions resides in the ability for NAP to be trained end-to-end for acquisition function value prediction. At test time, we show that this end-to-end training allows to achieve better performance, making our contributions significant.
>
> >Results were only included on four tasks.
>
> Regarding the concerns about experimental evidence, we believe that we show the performance of NAP on a panel of experiments, ranging from hyperparameter optimisation in low dimension (HPO-B) to high dimension (MIP) and from continuous spaces (HPO-B) to combinatorial spaces (Antibody, EDA) and even mixed-type spaces (MIP). This gives, in our opinion, a good overview of the practical applications where BO in general can be used and where meta-BO methods such as NAP can also be used. Please also note that the **HPO-B experiment is actually a collection of 6 tasks** (see appendix) and is an established benchmark for meta-learning [5,6].
>
> >How were the hyperparameters chosen for the implementation if no hyperparameter search was conducted.
>
> Thank you for raising the topic of hyperparameters.
> We did not perform any hyperparameter tuning on the values provided in the table in appendix an in particular we did not tune the weight between the two losses, which is simply set to 1.0 in all experiments. Moreover, all hyperparameters are fixed across all the experiments for a fair comparison. We set the hyperparameters based on the values from the codebase of MetaBO [2] and PFN [3] because we consider that their values have been optimised by their authors.
> Finally, with this single set of hyperparameters, we were able to obtain better results on all experiments, compared to SOTA-baselines (FSBO [6], OptFormer-EI [5], GP-EI).
>
> >Somewhat limited experimental evidence since no hyperparameter search was conducted on the proposed method or baseline.
>
> We do not conduct a full hyperparameter search for the reasons mentioned above, however we conduct an ablation study that can be seen as an analysis on the hyperparameter $\lambda$ giving the relative weights of the two losses in the total loss (see appendix C.1.). If the weight of the auxiliary loss is zero, we have NAP-RL and if the loss of the RL loss is zero, we have NP-EI. We can see that simply with equal weighting ($\lambda = 1$), we achieve better results.
>
> ----------
>
> [1] Bing-Jing Hsieh, Ping-Chun Hsieh, and Xi Liu. Reinforced few-shot acquisition function learning for bayesian optimization. In M. Ranzato, A. Beygelzimer, Y. Dauphin, P.S. Liang, and J. Wortman Vaughan, editors, Advances in Neural Information Processing Systems, volume 34, pages 7718–7731. Curran Associates, Inc., 2021.
>
> [2] Michael Volpp, Lukas P. Fröhlich, Kirsten Fischer, Andreas Doerr, Stefan Falkner, Frank Hutter, and Christian Daniel. Meta-learning acquisition functions for transfer learning in bayesian optimization. In 8th International Conference on Learning Representations, ICLR 2020, Addis Ababa, Ethiopia, April 26-30, 2020.
>
> [3] Samuel Müller, Noah Hollmann, Sebastian Pineda-Arango, Josif Grabocka, and Frank Hutter. Transformers can do bayesian inference. In The Tenth International Conference on Learning Representations, ICLR 2022, Virtual Event, April 25-29, 2022.
>
> [4] Tung Nguyen and Aditya Grover. Transformer neural processes: Uncertainty-aware meta learning via sequence modeling. In Kamalika Chaudhuri, Stefanie Jegelka, Le Song, Csaba Szepesvári, Gang Niu, and Sivan Sabato, editors, International Conference on Machine Learning, ICML 2022, 17-23 July 2022, Baltimore, Maryland, USA, volume 162 of Proceedings of Machine Learning Research, pages 16569–16594. PMLR, 2022.
>
> [5] Yutian Chen, Xingyou Song, Chansoo Lee, Zi Wang, Richard Zhang, David Dohan, Kazuya Kawakami, Greg Kochanski, Arnaud Doucet, Marc Aurelio Ranzato, Sagi Perel, and Nando de Freitas. Towards learning universal hyperparameter optimizers with transformers. In
> S. Koyejo, S. Mohamed, A. Agarwal, D. Belgrave, K. Cho, and A. Oh, editors, Advances in Neural Information Processing Systems, volume 35, pages 32053–32068. Curran Associates, Inc., 2022.
>
> [6] Martin Wistuba and Josif Grabocka. Few-shot bayesian optimization with deep kernel surrogates. In 9th International Conference on Learning Representations, ICLR 2021, Virtual Event, Austria, May 3-7, 2021.

---

> > ### Comment · Reviewer_P5Ji · 2023-08-19
> > **Raising Rating to 5**
> >
> > I would like to thank the authors for responding to my questions and for their clarifications. After reviewing their response and the discussion with other authors I will be raising my Rating to a 5 due to the clarifications on hyperparameter search and E2E differentially.

---

### Official Review · Reviewer_DqJs · 2023-07-27

**Soundness:** 3 good
**Presentation:** 2 fair
**Contribution:** 3 good
**Rating:** 6
**Confidence:** 2

**Summary:**

By combining a Transformer-based architecture and reinforcement learning objective, this paper proposes the first end-to-end framework for meta-Bayesian optimization.
The proposed framework is claimed to resolve the inefficiency in the existing two-stage approaches.

**Strengths:**

- The meta-Bayesian optimization is an important topic that can be useful in various domains.
- The presented motivation appears to be compelling, while the proposition of employing a Transformer-based architecture seems reasonable.
- The experiments include practical problem settings.

**Weaknesses:**

- I am not sure if RL is really the optimal way to achieve an end-to-end meta-BO framework. I think this paper lacks a justification for why RL is a necessary component.
- Although this paper claims to tackle the inefficiencies from two-stage approaches, introducing RL makes training much harder. The authors add an auxiliary task to solve this issue, but due to this, the proposed method does not seem to reduce the complexity and the objective discrepancy in the previous two-stage approaches.

Please note that I do not have much expertise in the relevant fields.
I currently do not see any major flaw in this work, but I am also not fully convinced.
Comments from other reviewers and further discussion may largely change my score.

**Questions:**

- Is there any factor that necessitates the use of RL?

**Limitations:**

Yes

---

> ### Author Rebuttal · Authors · 2023-08-09
>
> We thank the reviewer for their review and for acknowledging the strengths of our work and the relevance of our experiments. We hope that we can further motivate the use of RL and auxiliary loss in our response below.
>
> >Is there any factor that necessitates the use of RL?
>
> To answer your first point, we would like to underline that the necessity of using RL comes from the fact that we are learning acquisition functions. As we do not have true labels for how good it is to select a point (it depnds on the other points seen so far and their values), we cannot rely on supervised learning and thus choose to rely on RL. The NP-EI baseline - trained only using a supervised loss and EI acquisition - shows that there is value in meta-learning the acquisition with RL as NAP outperforms it on every task.
>
> >Although this paper claims to tackle the inefficiencies from two-stage approaches, introducing RL makes training much harder. The authors add an auxiliary task to solve this issue, but due to this, the proposed method does not seem to reduce the complexity and the objective discrepancy in the previous two-stage approaches.
>
> Even in a two-stage approach we would need to train using RL for the second stage. Indeed, we can pretrain a neural process surrogate with supervised loss but to meta-learn a neural acquisition we must introduce RL. We add the auxiliary loss to help tackle the sparsity of the RL reward, which introduces a useful inductive bias. Even though this auxiliary loss is indeed not exactly optimising the same objective as in the downstream task, we argue that (see appendix C.1.) training end-to-end enables NAP to be updated simultaneously from both signals hence reducing the objective discrepancy. We use an end-to-end architecture so we can backpropagate information both from the acquisition (RL) and the auxiliary loss (supervised) through the whole network, which is not possible in a two-stage approach (see Figure 3 in appendix).

---

> > ### Comment · Reviewer_DqJs · 2023-08-16
> >
> > Thank you for the response.
> >
> > I'm satisfied with the response and raising the score from 5 to 6.

---

### Author Rebuttal · Authors · 2023-08-09

We thank all reviewers for their time reading our paper and writing reviews. We are very appreciative of all positive comments made about our work, on clarity, applicability of the method, quality of experiments and baselines implementation. We further thank all reviewers for raising interesting points and questions about motivation, novelty and hyperparameters. We address each of the reviews separately in their own threads but we summarize here the main remarks.

### Novelty

We discuss prior works that make use of transformers for BO as well as prior work using RL to learn acquisition functions. However, to the best of our knowledge, we are the first to propose an **end-to-end differentiable** method that specifically learns acquisition values with RL and is based on a transformer neural process architecture. We will clarify this claim in the main paper. We also discuss orthogonal approaches that learn a better prior from the source tasks data for pretraining, as well as using RL for learning acquisitions in order to do better adaptation (fine tuning) at test time. We will also extend our discussion of these research directions and stress their differences with ours.

### Hyperparameters

We will make it clearer in the main paper that the hyperparameters of NAP were not tuned. This is only fair as we did not tune them for the baselines either, considering that the ones set by the authors in their respective code repositories were already optimised. Furthermore, we used the same  values for all of the experiments, avoiding task-specific hyperparameter tuning that could limit generalisation and applicability of our method. In particular we used an equal weight between the RL loss and the auxiliary loss in all experiments ($\lambda = 1$). An ablation for that particular parameter has been carried out and results are shown in Appendix C.1.

### Extra experiment

We added the NP-EI baseline to the HPO-B experiment as noted by one reviewer, the update plots are in the PDF document added in this rebuttal.

---

### Decision · Program_Chairs · 2023-09-21

**Decision:**

Accept (poster)

**Comment:**

This paper proposes a meta-learning method of Bayesian optimization. With the proposed method, transformer-based neural processes are used to model acquisition functions, and they are trained in an end-to-end fashion based on reinforcement learning. The effectiveness of the proposed method is demonstrated in their extensive experiments using different types of tasks with proper baselines. Meta-Bayesian optimization is an important task. The design of the proposed method is reasonable and effective. The auxiliary loss used in the proposed method is interesting. This paper is well written. It would be good to add more discussion on related work on meta-learning of Bayesian optimization, e.g., End-to-End Learning of Deep Kernel Acquisition Functions for Bayesian Optimization, arXiv, 2111.00639, 2021, to clarify the contribution of this paper. The additional experimental results submitted in the author response strengthen the paper.